# Potential Tamoxifen Repurposing to Combat Infections by Multidrug-Resistant Gram-Negative Bacilli

**DOI:** 10.3390/ph14060507

**Published:** 2021-05-26

**Authors:** Andrea Miró-Canturri, Rafael Ayerbe-Algaba, Raquel del Toro, Manuel Enrique-Jiménez Mejías, Jerónimo Pachón, Younes Smani

**Affiliations:** 1Clinical Unit of Infectious Diseases, Microbiology and Preventive Medicine, University Hospital Virgen del Rocío, 41013 Seville, Spain; amirocan93@gmail.com (A.M.-C.); ayerberafael@gmail.com (R.A.-A.); mej-mejias@telefonica.net (M.E.-J.M.); 2Institute of Biomedicine of Seville (IBiS), University Hospital Virgen del Rocío, CSIC, University of Seville, 41013 Seville, Spain; rdeltoro-ibis@us.es (R.d.T.); pachon@us.es (J.P.); 3Department of Medical Physiology and Biophysics, Institute of Biomedicine of Seville, University of Seville, 41009 Seville, Spain; 4Department of Medicine, University of Seville, 41009 Seville, Spain

**Keywords:** repurposing drug, tamoxifen, bacteria, infection, animal model, immune system

## Abstract

The development of new strategic therapies for multidrug-resistant bacteria, like the use of non-antimicrobial approaches and/or drugs repurposed to be used as monotherapies or in combination with clinically relevant antibiotics, has become urgent. A therapeutic alternative for infections by multidrug-resistant Gram-negative bacilli (MDR-GNB) is immune system modulation to improve the infection clearance. We showed that immunocompetent mice pretreated with tamoxifen at 80 mg/kg/d for three days and infected with *Acinetobacter baumannii*, *Pseudomonas aeruginosa*, or *Escherichia coli* in peritoneal sepsis models showed reduced release of the monocyte chemotactic protein-1 (MCP-1) and its signaling pathway interleukin-18 (IL-18), and phosphorylated extracellular signal-regulated kinase 1/2 (ERK1/2). This reduction of MCP-1 induced the reduction of migration of inflammatory monocytes and neutrophils from the bone marrow to the blood. Indeed, pretreatment with tamoxifen in murine peritoneal sepsis models reduced the bacterial load in tissues and blood, and increased mice survival from 0% to 60–100%. Together, these data show that tamoxifen presents therapeutic efficacy against MDR *A. baumannii*, *P. aeruginosa*, and *E. coli* in experimental models of infection and may be a new candidate to be repurposed as a treatment for GNB infections.

## 1. Introduction

Infections caused by Gram-negative bacilli (GNB) such as *Acinetobacter baumannii*, *Pseudomonas aeruginosa*, and *Escherichia coli* represent an increasing worldwide problem. 

Sepsis is a major cause of mortality in acute infections, characterized by a systemic inflammatory response syndrome caused by them [1,2]. In the United States the incidence of sepsis was increased from 50–95 to 535 cases per 100,000 inhabitants between 2003 and 2015 [3,4]. Despite the correct management that includes adequate resuscitation and early antibiotic treatment, mortality in patients with sepsis and septic shock is very high, ranging between 25% and 70% [2,5].

In 2017, the World Health Organization listed these pathogens as the first antibiotic-resistant “priority pathogens” that pose the greatest threat to human health. There is, therefore, an urgent need to find new antimicrobial agents against extensive and pan-drug-resistant GNB. Two key approaches can help alleviate the problem of antibiotic resistance, first by targeting the bacterial virulence factors without inhibiting bacterial growth, which can slow the development of drug resistance by reducing the selective pressure on the bacteria [6,7] and, secondly, by the modulation/regulation of the immune system response to improve the infection development [8,9]. In this context, some studies focused on the stimulation of the immune system to treat bacterial infections using molecules including lysophosphatidylcholine as monotherapy and as adjuvant for the antimicrobial treatment [8,10,11] or 3′-5′-cyclic diguanylic acid (c-di-GMP), which increase neutrophils protecting against *A. baumannii* infection [12]. 

Inflammatory monocytes and neutrophils derived from bone marrow are important cellular mediators of the innate immune response against bacterial infections. During the early stages of a bacterial infection, both cell populations migrate from the bone marrow to the bloodstream and, subsequently, to the sites of infection [13,14]. This migration is partially regulated by the monocyte chemotactic protein-1 (MCP-1), whose expression is increased in bone marrow mesenchymal cells in response to circulating Toll-like receptor ligands and results in the mobilization of inflammatory monocytes [15]. It is well established that MCP-1 release is controlled by interleukin-18 (IL-18) and extracellular signal-regulated kinase 1/2 (ERK1/2) [16], and the levels of MCP-1 are higher in patients with sepsis, septic shock, and pneumonia [17,18]. 

It has been well documented that anticancer drugs like tamoxifen can modify the immune response by regulating cytokine release [19]. Mechanistically, tamoxifen has been reported to reduce MCP-1 transcription and expression in human coronary artery endothelial cells and endometrial cancer cells, respectively [20,21]. As MCP-1 is involved in the immune cells’ migration, it may be hypothesized that an undiscovered connection between MCP-1 release and immune cells’ migration after bacterial infection and treatment with tamoxifen is present.

In this study, we report that tamoxifen downregulates the expression of MCP-1, impairing the migration of bone marrow-derived cells to the bloodstream induced by *A. baumannii*, *P. aeruginosa*, and *E. coli* and, consequently, modulating the inflammatory response. In a murine peritoneal sepsis model, we observed that tamoxifen decreased the development of infection by these pathogens, lowering their concentrations in tissues and blood and increasing mice’s survival.

## 2. Results

### 2.1. Bone Marrow Immune Cells Migrate in Response to Monocyte Chemotactic Protein-1 (MCP-1) and Interleukin-18 (IL-18) during Bacterial Infection

To determine whether bacterial infection influences circulating immune cells from the bone marrow in response to MCP-1 and IL-18, a MCP-1 controller [16], we intraperitoneally (ip.) administered *A. baumannii*, *P. aeruginosa*, and *E. coli* to mice and measured the proportions of myeloid cells CD11b+, inflammatory monocytes CD11b+Ly6C^hi^, and neutrophils CD11b+Ly6G+. After 24 h of *A. baumannii* infection, the percentages of immune cells (myeloid cells, inflammatory monocytes, and neutrophils) were decreased in bone marrow and were increased in blood (Figure 1A–C). The same results were observed when mice were infected with *P. aeruginosa* and *E. coli* (Figure 1A–C). In the spleen, the percentages of these immune cells, from a total of 1.6 × 10^6^ cells/mL, were decreased less after infection with *A. baumannii*, *P. aeruginosa*, and *E. coli* for 24 h (Appendix A); indicating that the increase of circulating monocytes and neutrophils did not proceed from the splenic reservoir [22,23]. Interestingly, no significant differences in the population of lymphocytes B and T CD4+ and CD8+ were observed between non-infected and infected mice in the bone marrow, spleen, and blood 24 h post-bacterial infection, except for the bone marrow lymphocytes CD8+ of mice infected with *P. aeruginosa* (Appendix A).

A paradigm widely accepted is the formation of chemokine gradients to guide inflammatory cells to the sites of infection [24]. Among them, MCP-1 has been shown to be involved in the migration of immune cells from the bone marrow to the bloodstream after binding to the CCR2 receptor [25]. As shown in Figure 1D, mice infected with *A. baumannii*, *P. aeruginosa*, and *E. coli* for 6 and 24 h increased the release of MCP-1 in mice serum significantly and progressively (between 1000 and 4000 µg/mL). It is well known that MCP-1 release is controlled by IL-18 and ERK1/2 [16]. Consequently, the levels of IL-18 in mice serum gradually increased 6 and 24 h after infection with *A. baumannii*, *P. aeruginosa*, and *E. coli*. The IL-18 levels at 24 h were 2144 ± 408.1 µg/mL, 7286 ± 1056 µg/mL, and 3124 ± 671.3 µg/mL, respectively (Figure 1E). Moreover, ERK1/2 was phosphorylated 2 h after infection of the RAW 264.7 macrophage cell line in vitro with *A. baumannii*, *P. aeruginosa*, and *E. coli*, defining the activation of kinase response to these pathogens (Figure 1F). 

To determine whether MCP-1 is involved in the migration of inflammatory monocytes and neutrophils from bone marrow to blood, wild-type (WT) mice and mice lacking the MCP-1 protein (MCP-1 knockout (KO) mice) were infected with *A. baumannii*, *P. aeruginosa*, and *E. coli*. First, we detected MCP-1 release only in WT mice (Figure 2A). Importantly, the infection of MCP-1 KO mice with these pathogens showed that the migration of inflammatory monocytes and neutrophils from the bone marrow to the blood (Figure 2B,C) causes a reduction of 6.76 ± 1.55% to 2.17 ± 1.14% (*p* = 0.049) and from 12.67 ± 1.37% to 4.13 ± 0.99% (*p* = 0.043), respectively, for *A. baumannii* infection. Similar results were observed when MCP-1 KO mice were infected with *P. aeruginosa* and *E. coli* strains (Figure 2B,C). Non-infected WT and MCP-1 KO mice presented similar inflammatory monocytes and neutrophil proportions in the bone marrow (*p* = 0.85 and *p* = 0.71) and in blood (*p* = 0.34 for both cells), indicating that the lack of MCP-1 did not affect the migration of these cells from the bone marrow in basal conditions (Figure 2B,C). These data suggest that MCP-1 is involved in the traffic of immune cells from the bone marrow to the blood after infection with *A. baumannii*, *P. aeruginosa*, and *E. coli*.

### 2.2. Tamoxifen Impairs the Migration of Immune Cells from Bone Marrow to Blood

In order to study whether tamoxifen can modulate the inflammation generated by bacterial infections, we treated the RAW 264.7 macrophage cell line with tamoxifen for 24 h and infected the cells with *A. baumannii*, *P. aeruginosa*, or *E. coli* for 2 h. After this incubation, we determined the secretion of MCP-1 in the macrophage cells supernatant (enzyme-linked immunosorbent assay (ELISA)) and the phosphorylation of ERK in the macrophage cells by Western blot. The treatment with tamoxifen decreased the release of MCP-1 and the phosphorylation of ERK1/2 in macrophages infected with these pathogens, compared to macrophages without tamoxifen treatment (Figure 3A,B). To confirm these data in vivo, mice were treated ip. with three doses of 80 mg/kg/d of tamoxifen before the bacterial infection. Serum was collected 6 and 24 h post-bacterial infection. Figure 3C reveals that treatment with tamoxifen reduced MCP-1 levels when compared with *A. baumannii*-, *P. aeruginosa*-, or *E. coli*-infected and not treated groups. It is worth highlighting that IL-18 levels were also reduced after tamoxifen treatment of mice infected with *A. baumannii*, *P. aeruginosa*, and *E. coli* (Figure 3D).

In order to confirm whether tamoxifen treatment increases the proportions of myeloid cells, inflammatory monocytes, neutrophils and dendritic cells (Tip-DC) in the bone marrow and reduces them in the blood, we administered tamoxifen to mice before infection with *A. baumannii*, *P. aeruginosa*, and *E. coli* for 24 h. Flow cytometric analysis demonstrated that in the bone marrow of mice infected with these pathogens, the percentage of myeloid cells and monocytes was higher in those treated with tamoxifen than in those that remained untreated. In contrast, the percentage of myeloid cells and monocytes in the blood of mice treated with tamoxifen was lower than in those untreated with tamoxifen (Figure 4A,B). A similar pattern was observed for the neutrophils and Tip-DC of mice infected with *A. baumannii* and *P. aeruginosa* (for neutrophils); however, in those infected with *E. coli*, tamoxifen did not significantly increase the percentage of neutrophils in bone marrow when compared with the untreated group (Figure 4C,D). 

MCP-1 KO mice showed an impaired migration of inflammatory monocytes and neutrophils from the bone marrow to the blood after bacterial infection (Figure 2B,C). In order to determine whether tamoxifen is able to reduce this migration in mice deficient in MCP-1 secretions, we treated MCP-1 KO mice with tamoxifen and infected them with *A. baumannii*, *P. aeruginosa*, and *E. coli*. As shown in Figure 5A,B, tamoxifen-treated mice presented a reduction in the migration of inflammatory monocytes and neutrophils, despite the lack of MCP-1. Both populations were more present in the bone marrow, and concentrations in the blood were also reduced when compared with WT mice treated with tamoxifen and infected with these pathogens (except for inflammatory monocytes and neutrophils from mice infected by *P. aeruginosa* and *E. coli*, respectively), indicating that tamoxifen may regulate other chemokines and migration pathways involved in this phenomenon (Figure 5 and Appendix A). 

### 2.3. Tamoxifen Enhances Bacterial Killing by Macrophages and Neutrophils In Vitro

Recent studies reported that treatment with tamoxifen enhances neutrophil activity by increasing the neutrophil extracellular traps (NETosis) and induces changes in macrophages by inhibiting the expression of CD36 and PPARγ, thereby reducing atherosclerosis [26,27], but there are no data regarding the immune function of both cells treated with tamoxifen after a bacterial infection. To determine whether tamoxifen can increase the killing activity of macrophages and neutrophils, assays with the RAW 246.7 and HL-60 neutrophils cell lines, pretreated with tamoxifen and infected with *A. baumannii*, *P. aeruginosa*, and *E. coli*, were performed. We demonstrated that macrophage incubation with tamoxifen (2.5 mg/L) for 2 or 6 h, followed by infection with *A. baumannii* for 2 h, decreased the amount of bacteria within the macrophage by 10% and 30%, respectively (Figure 6A), without affecting the amount of *A. baumannii* in the extracellular medium (Figure 6B). This suggests the increase of bacterial killing inside the macrophage after exposure to tamoxifen. Similar results were observed after treatment with tamoxifen and infection with *E. coli*, but not with *P. aeruginosa* (Figure 6A). Regarding neutrophil activity, incubation with 2.5 mg/L of tamoxifen for 2 or 6 h, followed by infection with *A. baumannii* for 2 h, increased bacterial killing by 5% and 25%, respectively. Similar results were observed after treatment with tamoxifen and infection with *E. coli*, but not with *P. aeruginosa* (Figure 6A). Accordingly, tamoxifen treatment increases the killing activity of macrophages and neutrophils against *A. baumannii* and *E. coli* but not against *P. aeruginosa*.

### 2.4. Tamoxifen Increases Mice Survival and Decreases the Bacterial Burden in a Murine Sepsis Model of A. Baumannii, P. Aeruginosa, and E. Coli

Our results demonstrated that tamoxifen plays an important role in innate immune cells’ trafficking after bacterial infection. Going further, we wanted to know whether tamoxifen could protect the mice against a lethal bacterial inoculum. We treated mice with tamoxifen (80 mg/kg/d), administered intraperitoneally for three days before the infection with a minimal lethal dose 100 (MLD100) of *A. baumannii*, *P. aeruginosa*, and *E. coli* and we monitored the mice survival for three days. Pretreatment with tamoxifen increased the mice’s survival after infection with *A. baumannii*, *P. aeruginosa*, and *E. coli* to 100%, 66.7%, and 83.3% (*p* < 0.01), respectively (Table 1). Table 1 shows that treatment with tamoxifen decreased the spleen and lung bacterial concentrations of these pathogens by 6.64 and 7.16 log_10_ colony-forming units (CFU)/g (*p* < 0.015; for *A. baumannii*), by 3.58 and 5.1 log_10_ CFU/g (*p* < 0.015; for *P. aeruginosa*), and by 3.7 and 4.16 log10 CFU/g (*p* < 0.015; for *E. coli*), compared with the control infected groups. Blood bacterial concentrations presented a decrease compared to control infected groups of 5.53, 5.45, and 4.31 log_10_ CFU/mL (*p* < 0.01) for *A. baumannii*, *P. aeruginosa*, and *E. coli*, respectively. Similar efficacy of tamoxifen was observed in a murine peritoneal sepsis model by susceptible and MDR clinical isolates of *A. baumannii*, *P. aeruginosa*, and *E. coli*. Treatment with tamoxifen increased the mice’s survival to 66.7%, 83.3%, and 50% (*p* < 0.01) for non-MDR *A. baumannii*, *P. aeruginosa*, and *E. coli*, respectively, and 83.3%, 66.7%, and 50% (*p* < 0.01) for the MDR *A. baumannii*, *P. aeruginosa*, and *E. coli* harboring the *mcr-1* gene (Figure 7). These findings indicate that tamoxifen treatment presents good therapeutic efficacy against reference and clinical isolates of *A. baumannii*, *P. aeruginosa*, and *E. coli*.

### 2.5. Direct Treatment with Tamoxifen Increases Mice Survival and Decreases the Bacterial Burden in a Murine Sepsis Model of A. Baumannii

In order to demonstrate the efficacy of tamoxifen in mice already infected with *A. baumannii* we have used the recommend dose of 250 mg/kg by Corriden et al. 2015 [26] administered in 6 infected mice 2 h post-bacterial inoculation. We found that tamoxifen treatment reduced the bacterial loads in spleen, lungs and blood to 5.37 ± 1.43 log CFU/g (*p* = 0.02), 6.72 ± 0.62 log CFU/g (*p* = 0.0007) and 3.74 ± 1.29 log CFU/mL (*p* = 0.33), respectively, and increased mice survival to 50% (*p* = 0.01).

## 3. Discussion

The present study provides new data highlighting the antibacterial effect of tamoxifen. Here, we provide the first evidence of an essential role played by tamoxifen in the regulation of immune cells’ traffic after bacterial infection, in order to reduce the hyperinflammation caused by sepsis.

This study, as well as previous works [13,14], showed that the regulation of inflammatory monocytes’ and neutrophils’ migration is important in the host defense against bacterial infections. This is consistent with the immune system modulation that improves the bacterial infection clearance [9]. Exploiting immunomodulatory drugs approved by the regulatory agencies for clinical indications different to bacterial infection therapy has several advantages [28], including the fact that their pharmacological characteristics (toxicity and pharmacokinetics) in preclinical and clinical trials are available. Therefore, the time and economic costs of the evaluation of these drugs in other therapeutic applications, such as the treatment of bacterial infections, will be reduced [29]. 

Here, we showed that tamoxifen reduces the release of MCP-1 and IL-18, and the phosphorylation of ERK. We suggest that the reduction of IL-18 secretion by tamoxifen may drive the reduction of MCP-1 release through a reduction of ERK phosphorylation, which would contribute to efficient reduction of the migration of inflammatory monocytes and neutrophils from the bone marrow to the blood. It is well known that, during sepsis, bacterial infection releases high levels of harmful substances, resulting in the activation of systemic immune response and the development of hyperinflammation [30,31]. Thus, the decrease of certain immune cells’ migration to the bloodstream and their activation may reduce the hyperinflammation state and the secondary deleterious effects observed in sepsis.

Recruitment of immune cells from the bone marrow to the blood during systemic infection with GNB is probably mediated by multiple pathways dependent on or independent of MCP-1, such as MyD88 and MIP-2 [32,33,34]. MyD88 has been reported to induce MCP-1 release of macrophages after their infection with *Listeria monocytogenes* [32]. In contrast, to our knowledge, MIP-2 is not involved in the release of MCP-1 by eukaryotic ells. The presence of pathways independent of MCP-1 has been confirmed in a study of MCP-1 KO mice infected with *A. baumannii*, *P. aeruginosa*, and *E. coli*, in which inflammatory monocytes and neutrophils migrate at lower levels from the bone marrow to the bloodstream. Previous independent work reported that the deletion of MCP-1 in mice did not completely abolish the recruitment of monocytes during infection with *L. monocytogenes*; this recruitment was diminished by 40–50% [35], suggesting the involvement of MCP-3, another monocyte chemoattractant protein, after binding to the CCR2 receptor in the systemic bacterial infection [36]. Regarding neutrophils, although it is widely accepted that MIP-2 stimulates their migration from the bone marrow [37,38], we demonstrated for the first time that in MCP-1 KO mice the migration of neutrophils from the bone marrow to the bloodstream after GNB infection was diminished, suggesting the involvement of MCP-1 in this process. This result is consistent with the previous observation that MCP-1 regulates the recruitment of neutrophils to the lungs after *E. coli* infection [34]. Based on these data, MCP-1 plays an important role in the migration of inflammatory monocytes and neutrophils from the bone marrow to the bloodstream. However, this migration in MCP-1 KO mice infected with GNB and treated with tamoxifen is reduced but not abolished. A possible explanation could be the involvement of other MCP-1-independent pathways regulated by tamoxifen. In this context, further studies are required to decipher the role of these MCP-1-independent pathways in this process.

A consequence of the reduction in monocyte proportions in the blood after treatment with tamoxifen would be the reduction of macrophages and dendritic cells in the blood and tissues. Although the number of macrophages and neutrophils recruited to the sites of infection in mice treated with tamoxifen was lower, our in vitro assays suggested that their killing activity against *A. baumannii* and *E. coli* was enhanced by tamoxifen. The inflammatory monocytes are the precursors of a subset of dendritic cells (Tip-DC) that produce tumor necrosis factor-α (TNF-α) and inducible oxide synthase (iNOS), contributing to the innate defense against *L. monocytogenes* infection [39,40]. In contrast, another study reported that the reduction of proinflammatory monocytes and Tip-DC during *Trypanosoma brucei* infection diminished their pathogenicity [41]. These contradictory results in terms of the effect of monocytes and Tip-DC recruitment on host survival could be explained by the difference in cellular location of each pathogen: *L. monocytogenes* is intracellular, whereas *T. brucei* remains in the plasma [14]. Moreover, it is reported that tamoxifen inhibits the maturation of TipDC in vitro, in the presence of 17 β-estradiol, which does not respond sufficiently to bacterial LPS [42]. We suggest that the reduction in the dendritic cells’ proportions, along with their stunted maturation after tamoxifen treatment, produced a reduction in TNF-α and iNOS production, minimizing their deleterious effects in sepsis situations. Of note, in our study, we found that treatment with tamoxifen reduced the proportions of Tip-DC in blood and the release of proinflammatory cytokines such as TNF-α and IL-6 (Figure 4D, Appendix A). Accordingly, although we previously pointed out that *A. baumannii* could support intracellular life [43,44], the bacterial species used in our study are viewed as extracellular pathogens and are present in the blood. Consequently, it is possible that in our model, the reduction of monocyte and Tip-DC concentration by tamoxifen treatment, and the reduction of proinflammatory cytokines’ release, may play an important role in the therapeutic efficacy of tamoxifen.

In this study, differences were observed when *P. aeruginosa* was the infecting pathogen of macrophages versus *E. coli* and *A. baumannii* (Figure 6). We suggest that the differences observed when *P. aeruginosa* is the infecting agent versus *E. coli* and *A. baumannii* is due to the virulence factors present in *P. aeruginosa*. Previous reports showed that *P. aeruginosa* was less ingested by macrophages than *E. coli* [45]. *P. aeruginosa* enters and survives inside primary human and murine macrophages. This entrance depends on the expression of OprF, the most abundant bacterial outer membrane protein (OMP)/porin, the type III secretion system, and the exoS [46,47]. In addition, other authors have demonstrated that murine macrophages are more sensitive to the type III secretion system of *P. aeruginosa* than that of *E. coli* [48]. Regarding the comparison between *P. aeruginosa* and *A. baumannii*, a similar pattern has been observed with *E. coli*. *P. aeruginosa* persists more than *A. baumannii* inside normoxic and hypoxic murine macrophages [49]. Additional studies are needed to understand how tamoxifen increases the macrophages and neutrophils killing of *A. baumannii* and *E. coli*, and which mechanism used by *P. aeruginosa* to avoid this killing.

Tamoxifen dosage used in this study was 1.6 mg/d for three days. Only one study has reported in vivo the beneficial effect of tamoxifen against infection by Gram-positive bacterium *Staphylococcus aureus* [26]. They used a non-toxic dose of tamoxifen at 250 mg/kg (5 mg/d for 1 day) to treat mice infected by *S. aureus* [26]. We have analysed the toxicity effect of tamoxifen in mice at lower and higher doses of tamoxifen: 80 mg/kg (1.6 mg/d, the dosage used in our preventive experiments), 160 mg/kg (3.2 mg/d) and 320 mg/kg (6.4 mg/d), for three days by monitoring the mice survival for 7 days. We found that tamoxifen at 1.6 and 3.2 mg/d did not affect the mice survival being 100%, and tamoxifen at 6.4 mg/d reduced the mice survival to 83.33%. The cumulative doses of tamoxifen at 1.6 mg/d for three days of treatment correspond to 4.8 mg/d which is below to 5 mg/d in the study performed by Corriden et al. [26], and below the cumulative doses of 9.6 mg/d which did not affect the mice survival in our toxicity studies.

Data on the therapeutic efficacy of high tamoxifen dosages against GNB infections are not available. The next issue that must be addressed is to determine (i) the therapeutic efficacy of high tamoxifen dosages against *A. baumannii*, *P. aeruginosa* and *E. coli infections*, and (ii) whether tamoxifen given as direct treatment after bacterial infections will have an impact on bacterial burdens and mice survival against *P. aeruginosa* and *E. coli*, as a first step before evaluation in clinical studies. In addition, we need to determine the pharmacokinetic parameters of tamoxifen at 80/mg/kg ip. for 3 days in order to compare with those achieved in humans receiving tamoxifen at 40 mg/d po. for breast cancer treatment.

## 4. Materials and Methods

### 4.1. Reagents

Tamoxifen, porcine mucin, and protease inhibitors were obtained from Sigma, Spain.

### 4.2. Bacterial Strains

Reference *A. baumannii* ATCC 17978 [50], *P. aeruginosa* PAO1 [51] and *E. coli* ATCC 25922 [52] strains were used. We also used two clinical susceptible (Ab9) and multidrug-resistant (MDR) (Ab186) *A. baumannii* from the REIPI-GEIH 2010 collection [10], two clinical susceptible (Pa39) and MDR (Pa238) *P. aeruginosa* from the REIPI-GEIH 2008 collection [53], and two clinical susceptible (C1-7-LE) and MDR (EcMCR+, carrying *mcr-1* gene) *E. coli* [54,55]. 

### 4.3. Animals

Immunocompetent C57BL/6 female mice (16–18 g) were obtained from the University of Seville. MCP-1 KO mice were generated with a C57BL/6 background and obtained from Jackson Laboratory, USA. All mice were pathogen-free, were assessed for genetic authenticity, and were housed in regulation cages with food and water available ad libitum. This study was carried out in strict accordance with the protocol approved by the Committee on the Ethics of Animal Experiments of the University Hospital of Virgen del Rocío, Seville (0704-N-18). All surgery was performed under sodium thiopental anesthesia and all efforts were made to minimize suffering.

### 4.4. A. Baumannii, P. Aeruginosa, and E. Coli Peritoneal Sepsis Models

Murine peritoneal sepsis models with *A. baumannii*, *P. aeruginosa*, or *E. coli* strains were established by ip. inoculation of the bacteria in immunocompetent mice [7]. Briefly, six mice from each group were inoculated with the minimal bacterial lethal dose 100 (MLD100) of the bacterial suspensions, mixed in a 1:1 ratio with a saline solution containing 10% (*w/v*) porcine mucin. The MLD100 of ATCC 17978, Ab9, Ab186, PAO1, Pa39, Pa238, ATCC 25922, C1-7-LE, and EcMCR-1+ were 3.2, 5.9, 5.0, 4.9, 3.85, 6.7, 4.7, 2.91, and 6 log_10_ CFU/mL, respectively. Mortality was recorded over three or seven days. After the death or sacrifice of the mice at the end of the experimental period, aseptic thoracotomies were performed, and blood samples were obtained by cardiac puncture. The spleen and lungs were aseptically removed and homogenized (Stomacher 80 Biomaster; Seward, London, England) in 2 mL of sterile NaCl 0.9% solution. Ten-fold dilutions of the homogenized spleen, lungs and blood were plated onto sheep blood agar (Becton Dickinson Microbiology Systems, USA) for quantitative cultures. If no growth was observed after plating the whole residue of the homogenized tissue and blood, a logarithm value corresponding to the limit of detection of the method (1 CFU) was assigned. 

### 4.5. Therapeutic Effect of Tamoxifen in Immunocompetent Murine Models of Peritoneal Sepsis

The immunocompetent murine peritoneal sepsis models by *A. baumannii* (ATCC 17978, Ab9 and Ab186), *P. aeruginosa* (PAO1, Pa39 and Pa238), or *E. coli* (ATCC 25922, C1-7-LE and EcMCR-1+) strains were established by ip. inoculation of the bacteria in immunocompetent mice. Briefly, six animals from each group were infected ip. with 0.5 mL of the MLD100 of each strain mixed 1:1 with 10% porcine mucin. Tamoxifen therapy was administered for three days at one safe dose of 80 mg/kg/d, using corn oil as a vehicle [26], before bacterial inoculation. Mice were randomly assigned to the following groups: (1) controls (receiving corn oil as vehicle control), and (2) Tamoxifen administered at 80 mg/kg/d ip. for three days before bacterial inoculation with each strain. 

In addition, we have treated other six mice with tamoxifen at one dose of 250 mg/kg 2 h post-inoculation with MLD100 of *A. baumannii* ATCC 17978 strains. Mortality and bacterial loads in tissues and blood were determined, as described in a previous section. 

### 4.6. Flow Cytometry

Blood and bone marrow cells and spleen samples were prepared from control and infected mice with DML100 of ATCC 17978, PAO1, and ATCC 25922 strains and pretreated or not with tamoxifen, as described before [8,56]. Briefly, blood samples were harvested from mouse periorbital plexuses and resuspended in Ethylenediaminetetraacetic acid (EDTA). Bone marrow cells were collected by flushing mouse femurs with ice-cold phosphate-buffered saline (PBS), and red blood cells were depleted by lysis in a lysis buffer containing 0.15 M NH_4_Cl, 0.01 M KHCO_3_, and 0.01 M of disodium-EDTA. Spleen samples were harvested, homogenized, and filtered through a 20-µm mesh prior to red blood cells’ depletion by a lysis buffer. Bone marrow and spleen samples were lysed by a lysis buffer for 10 min at 4 °C, whereas blood samples were lysed for 10 min at room temperature. Cells (1–2 × 10^6^ cells per sample) were resuspended in PBS with 2% fetal bovine serum (FBS) and then incubated with the appropriate dilution of antibody conjugates. Samples were analyzed with LSRFortessa Flow Cytometer (BD Biosciences, Franklin Lakes, NJ, USA) and the data obtained were analyzed with DIVA software (BD Biosciences). The staining protocols included a combination of the following antibodies: anti-CD11b (clone M1/70, Allophycocyanin (APC)), anti-Ly6C (clone, AL-21, fluorescein isothiocyanate (FITC)), anti-Ly6G (clone 1A8, PE), anti-CD4 (Clone GK1.5, PE), anti-CD8 (Clone 53-6.7, PE-Cy7), anti-CD11c (clone N418, BV421) and anti-I-A/I-E (clone M5/114.15.2) (BD Biosciences, Spain), and anti-CD19 (Clone 6D5, Brilliant Violet 421) (BioLegend, Spain) diluted at 1:300 in PBS with 2% fetal bovine serum.

### 4.7. Cytokine Assays

Blood samples were collected from the periorbital plexuses of mice infected with DML100 of ATCC 17978, PAO1, and ATCC 25922 strains and pretreated or not with tamoxifen, as previously described [11]. Serum levels of murine MCP-1, IL-6, IL-18, and TNF-α were collected 6 and 24 h post-bacterial infection, without or with tamoxifen treatment. MCP-1, IL-6, IL-18, and TNF-α levels were determined by ELISA kit (ThermoFisher, for MCP-1) and (Affymetrix eBioscience, for IL-6, IL-18, and TNF-α) in accordance with the manufacturer’s instructions. Furthermore, the extracellular medium of RAW 264.7 macrophage cells infected with 8 log_10_ CFU/mL of ATCC 17978, PAO1, and ATCC 25922, and previously pre-incubated or not with 2.5 mg/L tamoxifen for 24 h, was collected to determine the MCP-1 levels.

### 4.8. Cell Culture and Infection

Macrophage cell line RAW 264.7 was obtained from the American Type Culture Collection (LGC, UK) and grown in Dulbecco’s modified Eagle medium (DMEM, Invitrogen, Spain) supplemented with 10% heat-inactivated fetal bovine serum, vancomycin (50 mg/L), gentamicin (20 mg/L), amphotericin B (0.25 mg/L) (Invitrogen), and 1% HEPES (Invitrogen) in a humidified incubator, 5% CO_2_ at 37 °C, as described previously [49]. RAW 264.7 cells were routinely passaged every 3–4 days. The cells were seeded for 24 h in 24-well plates before bacterial infection for Western blot and adhesion assays.

Human promyelocytic leukemia HL-60 cells (ATCC CCL-240) were grown in RPMI 1640 medium (GE Healthcare Life Sciences, Spain) supplemented with 10% heat-inactivated fetal bovine serum, 2 mM l-glutamine (Sigma, Spain), 25 mM HEPES, and a penicillin/streptomycin solution in a humidified incubator, 5% CO_2_ at 37 °C. HL-60 cells’ culture and differentiation were performed as previously described with minor changes [57]. HL-60 cells were incubated with 1.3% DMSO (Sigma, Spain) for seven days and their differentiation into neutrophils (referred to as HL-60 neutrophils) was monitored by flow cytometry analysis, using anti-CD11b (clone ICRF44, APC) (BioLegend) and DAPI as viability markers. 

### 4.9. Western Blot Immunoblotting

Proteins of infected RAW 264.7 macrophage cells with 8 log_10_ CFU/mL of ATCC 17978, PAO1 and ATCC 25922, preincubated or not with 2.5 mg/L tamoxifen, were collected, homogenized in RIPA buffer supplemented with 1 mM phenylmethylsulfonyl fluoride (PMSF) and a 10% cocktail of protease inhibitors, and centrifuged at 13,000× *g* for 20 min at 4 °C. The supernatant was removed and the amount of proteins was determined using the BCA assay (Promega, Spain). The samples were stored at −80 °C until later use. Six micrograms of proteins of each sample were mixed with an equal volume of 2× Laemmli buffer, denatured by heating the mixture for 5 min at 95 °C, and then resolved by 4–15% sodium dodecyl sulfate polyacrylamide gel electrophoresi (SDS-PAGE). The separated proteins were transferred using polyvinylidene fluoride (PVDF) membranes (Amersham Bioscience, Spain), and the membranes were blocked for 2 h with PBS and 0.1% (*v*/*v*) Tween 20 (PBST buffer) containing 5% (*w*/*v*) milk. The PVDF membranes were then incubated overnight at 4 °C with primary antibody: rabbit anti-mouse p44/42 MAPK (Erk1/2) (1:1000 dilution), Phospho-p44/42 MAPK (Erk1/2) (1:1000 dilution) and β-actin (1:1000 dilution) (Cell Signaling Technology, Spain) diluted in a PBST buffer containing 5% milk. After washing with PBST buffer, the membranes were incubated for 1 h at room temperature with a horseradish peroxidase-conjugated donkey anti-rabbit IgG as secondary antibody (GE Healthcare, Spain) (dilution 1:2000) diluted in a PBST buffer containing 5% milk. Subsequently, immunoreactive proteins were visualized using the enhanced chemiluminescence protocol (Super ECL, Thermo Scientific, USA).

### 4.10. Macrophages Adhesion Assay

RAW 264.7 cells were pretreated with 2.5 mg/L tamoxifen for 2, 6, or 24 h; and infected with ATCC 17978, PAO1, and ATCC 25922 strains (MOI 1:100) for 2 h with 5% CO_2_ at 37 °C. Subsequently, infected RAW 264.7 macrophage cells were washed five times with prewarmed PBS and lysed with 0.5% Triton X-100. Diluted lysates were plated onto sheep’s blood agar and incubated at 37 °C for 24 h for the enumeration of developed colonies and then the determination of the number of bacteria that attached to and invaded RAW 264.7 cells. Alternatively, we determined the concentration of the extracellular medium bacteria after 6 h of tamoxifen incubation by plating diluted extracellular medium onto sheep’s blood agar. 

### 4.11. Neutrophil Killing Assay

HL-60 neutrophils were pretreated with 2.5 mg/L for 2 or 6 h; and infected with ATCC 17978, PAO1, and ATCC 25922 strains (MOI 1:100) for 2 h with 5% CO_2_ at 37 °C. Subsequently, HL-60 neutrophils were washed five times with prewarmed PBS by centrifugation and lysed with 0.5% Triton X-100. Diluted lysates were plated onto sheep’s blood agar and incubated at 37 °C for 24 h for the enumeration of developed colonies and then the determination of the number of bacteria. The neutrophil killing index was calculated according to the formula [(CFU in the absence of neutrophils—CFU in the presence of neutrophils)/CFU in the absence of neutrophils] × 100 [57]. 

### 4.12. Statistical Analysis

Group data are presented as means ± standard errors of the means (SEM). For in vitro studies, Student’s *t*-test was used to determine the differences between means. Differences in bacterial spleen, lung, and blood concentrations (mean ± SEM log_10_ CFU per g or mL) were assessed by analysis of variance (ANOVA) and post hoc Dunnett’s and Tukey’s tests. Differences in mortality (%) between groups were compared by use of the *χ*^2^ test. *p* values of <0.05 were considered significant. The SPSS (version 21.0; SPSS Inc.) statistical package was used.

## 5. Conclusions

Together, these data suggest that treatment with tamoxifen may be useful as prophylaxis in hospitalized patients with a risk of infection with Gram-negative bacilli.

## Figures and Tables

**Figure 1 pharmaceuticals-14-00507-f001:**
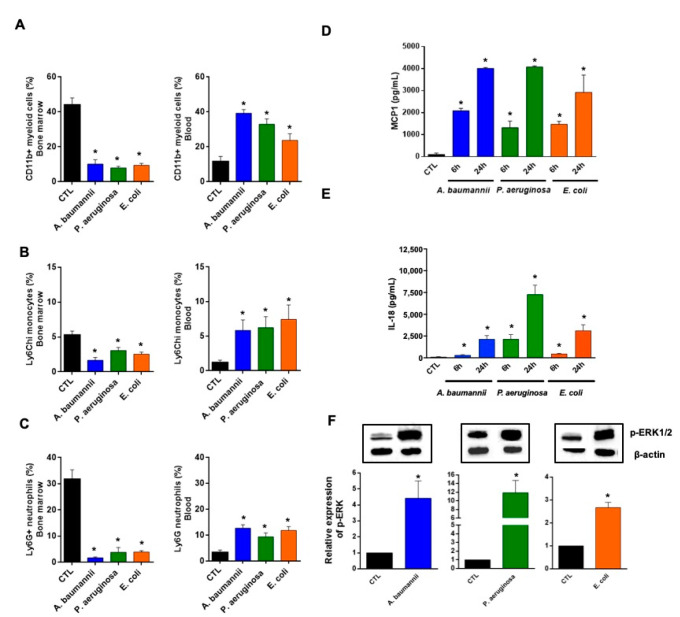
Bone marrow immune cells’ migration to the blood in response to monocyte chemotactic protein-1 (MCP-1) and interleukin-18 (IL-18) during bacterial infection. (**A**) Myeloid cells, (**B**) inflammatory monocytes, and (**C**) neutrophils were identified as CD11b+, CD11b+Ly6C^hi^, and CD11b+Ly6G+, respectively, by flow cytometry in the bone marrow and blood of mice infected with MLD100 of *A. baumannii* ATCC17978, *P. aeruginosa* PAO1, or *E. coli* ATCC25922 strains for 24 h. (**D,E**) Serum MCP-1 and IL-18 levels (enzyme-linked immunosorbent assays (ELISAs)), 6 and 24 h post-infection, in mice infected with minimal lethal dose 100 (MLD100) of *A. baumannii* ATCC17978, *P. aeruginosa* PAO1, or *E. coli* ATCC25922 strains. (**F**) RAW 264.7 cells were infected with *A. baumannii* ATCC17978, *P. aeruginosa* PAO1, or *E. coli* ATCC25922 strains for 2 h and proteins were collected for phospho-p44/42 Mitogen-Activated Protein Kinases (extracellular signal-regulated kinase 1/2 (Erk1/2)) and β-actin immunoblotting. Data are representative of six mice per group, and expressed as mean ± standard error of the mean (SEM). * *p* < 0.05: infected vs. control (CTL). CTL: non-infected mice, %: the percentage of myeloid cells, inflammatory monocytes and neutrophils from total cells in bone marrow or blood.

**Figure 2 pharmaceuticals-14-00507-f002:**
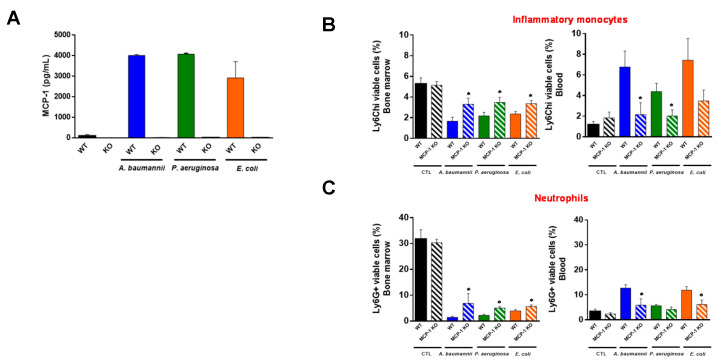
Role of MCP-1 in the bone marrow immune cells’ migration to the blood during bacterial infection. (**A**) Wild-type (WT) and MCP-1 knockout (KO) mice were infected with minimal lethal dose 100 (MLD100) of *A. baumannii* ATCC17978, *P. aeruginosa* PAO1, or *E. coli* ATCC25922 strains. Twenty-four hours post-infection, serum was harvested for MCP-1 ELISAs. (**B**) Inflammatory monocytes and (**C**) neutrophils were identified as CD11b+Ly6C^hi^ and CD11b+Ly6G+ by flow cytometry, respectively, in the bone marrow and blood of wild-type and MCP-1 KO mice infected with MLD100 of *A. baumannii* ATCC17978, *P. aeruginosa* PAO1, or *E. coli* ATCC25922 strains for 24 h. Data are representative of six mice per group, and expressed as mean ± SEM. * *p* < 0.05: WT vs. MCP-1 KO. WT: wild-type, MCP-1 KO: mice lacking MCP-1, CTL: non-infected mice, %: the percentage of inflammatory monocytes and neutrophils from total cells in bone marrow or blood.

**Figure 3 pharmaceuticals-14-00507-f003:**
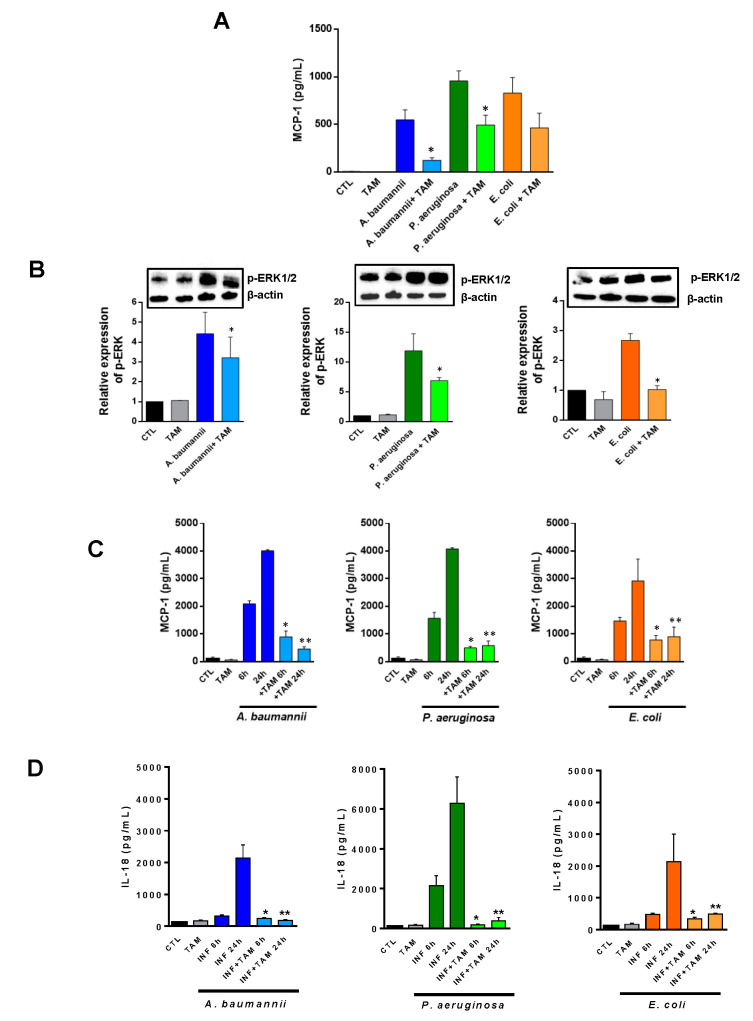
Tamoxifen reduces, after bacterial infection, the release of MCP-1 and IL-18 in vitro and in vivo, and ERK phosphorylation in vitro. (**A**,**B**) RAW 264.7 cells were treated with 2.5 mg/L of tamoxifen for 24 h and infected with *A. baumannii* ATCC17978, *P. aeruginosa* PAO1, or *E. coli* ATCC25922 strains for 2 h. MCP-1 levels and ERK-phosphorylation were determined by ELISA and immunoblotting assays, respectively. Data are representative of three independent experiments, and expressed as mean ± SEM. (**C**,**D**) Mice received firstly tamoxifen (80 mg/kg/d, for three days), followed by infection with a minimal lethal dose 100 (MLD100) of *A. baumannii* ATCC17978, *P. aeruginosa* PAO1, or *E. coli* ATCC25922 strains. At 6 and 24 h post-infection, serum was harvested for MCP-1 and IL-18 ELISA assays. Data are representative of six mice per group and are expressed as mean ± SEM. * *p* < 0.05: treated vs. INF, ** *p* < 0.05: treated vs. CTL. CTL: non-infected mice. TAM: tamoxifen. INF: infected.

**Figure 4 pharmaceuticals-14-00507-f004:**
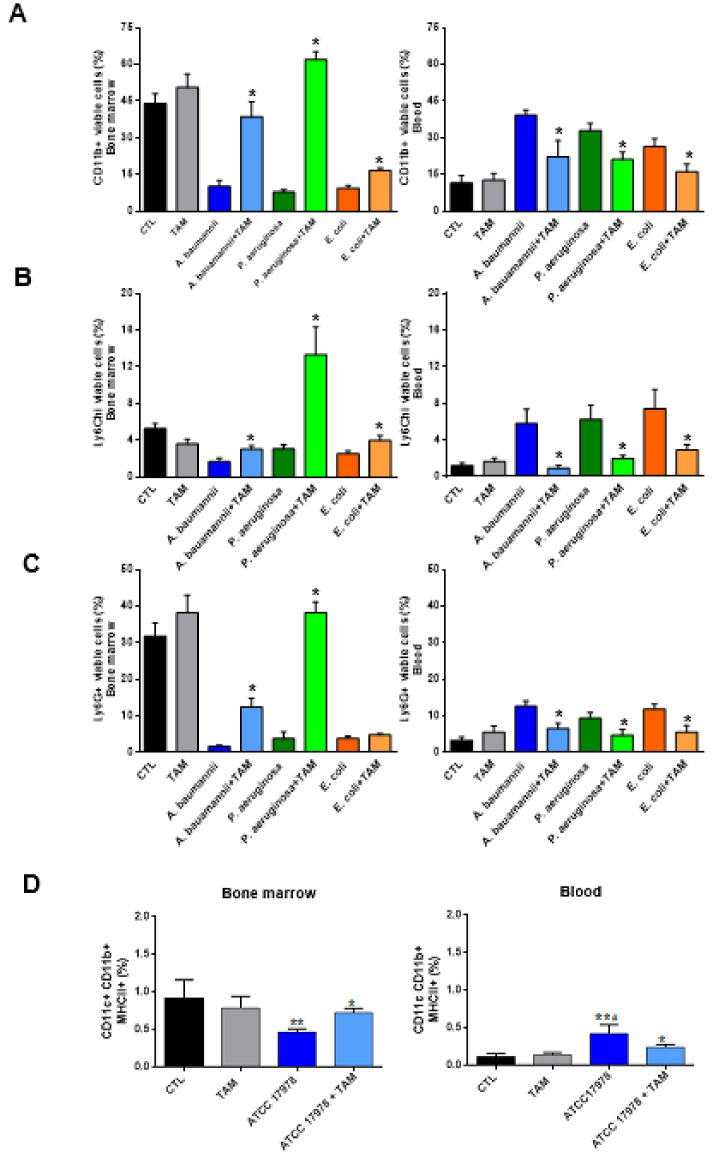
Tamoxifen impairs, after bacterial infection, the migration of immune cells from the bone marrow to the blood. Mice received firstly tamoxifen (80 mg/kg/d, for three days), followed by infection with minimal lethal dose 100 of *A. baumannii* ATCC17978, *P. aeruginosa* PAO1, or *E. coli* ATCC25922 strains. Twenty-four hours post-infection, (**A**) myeloid cells, (**B**) inflammatory monocytes, (**C**) neutrophils and (**D**) Tip-DC were identified as CD11b+, CD11b+Ly6C^hi^, CD11b+Ly6G+, and CD11b+CD11c+MHII+ respectively, by flow cytometry in the bone marrow and blood of mice. Data are representative of five/six mice per group and are expressed as mean ± SEM. * *p* < 0.05: treated vs. infected. ** *p* < 0.05: CTL vs. infected, ^#^
*p* < 0.05: TAM vs. infected. CTL: non-infected mice, TAM: tamoxifen, %: the percentage of myeloid cells, inflammatory monocytes, neutrophils and Tip-DC from total cells in bone marrow or blood.

**Figure 5 pharmaceuticals-14-00507-f005:**
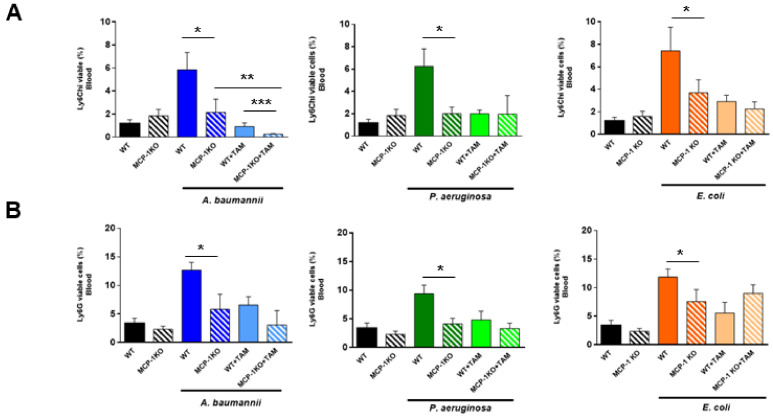
Tamoxifen impairs slightly, after bacterial infection in MCP-1-deficient mice, the migration of immune cells from the bone marrow to the blood through MCP-1 regulation. (**A**) WT and MCP-1 KO mice received firstly tamoxifen (80 mg/kg/d, for three days), followed by infection with a minimal lethal dose 100 of *A. baumannii* ATCC17978, *P. aeruginosa* PAO1, or *E. coli* ATCC25922 strains. Twenty-four hours post-infection, (**A**) inflammatory monocytes and (**B**) neutrophils were identified as CD11b+, CD11b+Ly6C^hi^, and CD11b+Ly6G+, respectively, by flow cytometry in the bone marrow and blood of mice. Data are representative of six mice per group and are expressed as mean ± SEM. CTL: non-infected mice. * *p* < 0.05: infected WT vs. infected MCP-1 KO, ** *p* < 0.05: infected MCP-1 KO vs. infected MCP-1 KO + TAM, *** *p* < 0.05: infected WT + TAM vs. infected MCP-1 KO + TAM. CTL: non-infected mice, TAM: tamoxifen, %: the percentage of inflammatory monocytes and neutrophils from total cells in blood.

**Figure 6 pharmaceuticals-14-00507-f006:**
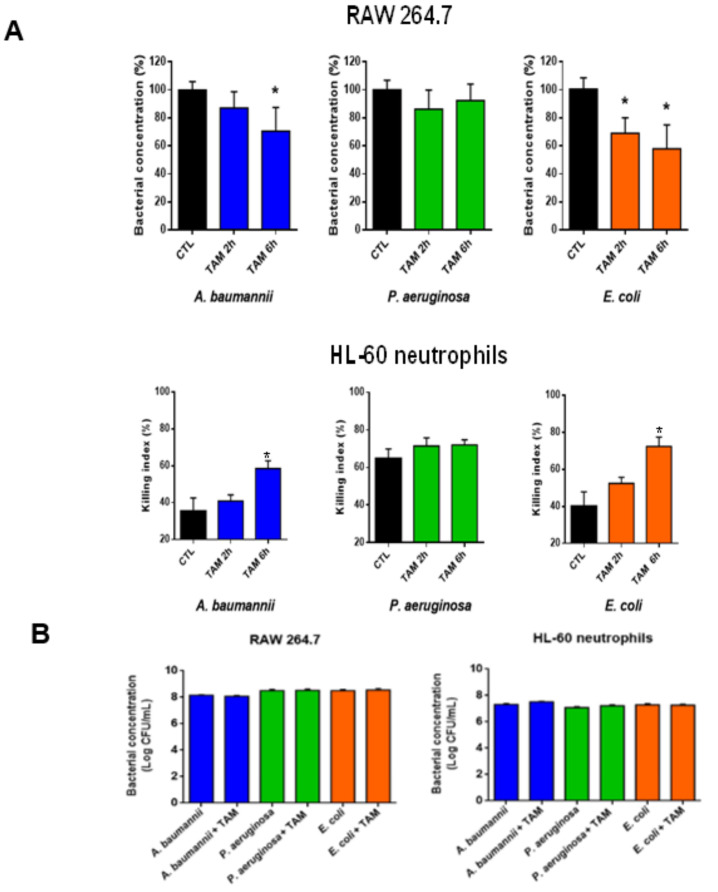
Tamoxifen induces the phagocytic activity of macrophages and neutrophils. RAW 264.7 cells and HL-60 neurophils were incubated with tamoxifen for 2 or 6 h and infected for 2 h with *A. baumannii* ATCC17978, *P. aeruginosa* PAO1, or *E. coli* ATCC25922 strains (MOI:100) (**A**) The adherence and invasion of these strains are expressed as the percentage of total control strain adhered or internalized to RAW 264.7 and the killing index of HL-60 neutrophil cells is expressed as the percentage. (**B**) The extracellular strains in the culture medium after 6 h of tamoxifen incubation are determined as Log colony-forming units (CFU)/mL. Data are representative of three independent experiments, and expressed as mean ± SEM. * *p* < 0.05: treated vs. CTL. CTL: infected cells without tamoxifen treatment. TAM: tamoxifen.

**Figure 7 pharmaceuticals-14-00507-f007:**
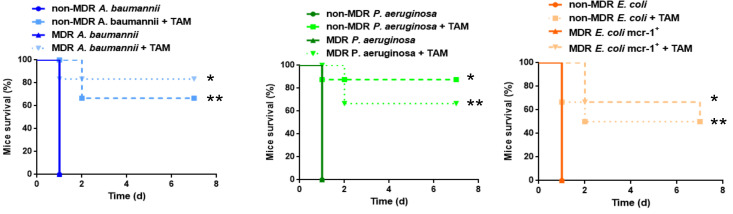
Tamoxifen increases the survival of mice infected with non-MDR or MDR GNB. Mice survival was monitored during 7 days for 6 mice infected with MLD100 of non-MDR and MDR *A. baumannii* (Ab9 and Ab186), *P. aeruginosa* (Pa39 and Pa238) or *E. coli* (C1-7-LE and EcMCR+) strains treated or not with 3 ip. doses of tamoxifen (80 mg/kg/d, for 3 days). * *p* < 0.05: treated vs. untreated, ** *p* < 0.05: treated vs. untreated, TAM: tamoxifen, MDR: multidrug-resistant.

**Table 1 pharmaceuticals-14-00507-t001:** Tamoxifen shows therapeutic efficacy in murine sepsis models by Gram-negative bacilli (GNB).

Strain	Treatment	*N*	Bacterial Load (log CFU/g or mL ± SEM)			3-Days Survival (%)
			Spleen	Lung	Blood	
*A. baumannii* ATCC 17978	CTL	6	9.51 ± 0.17	9.77 ± 0.17	6.14 ± 0.94	0
	TAM	6	2.87 ± 1.21 ^a^	2.61 ± 1.07 ^a^	0.61 ± 0.61 ^a^	100 ^a^
*P. aeruginosa* PAO1	CTL	5	8.91 ± 0.15	9.24 ± 0.17	6.71 ± 0.27	0
	TAM	6	5.33 ± 1.08 ^b^	4.14 ± 1.50 ^b^	1.26 ± 1.26 ^b^	66.7 ^b^
*E. coli* ATCC 25922	CTL	6	8.71 ± 0.05	8.88 ± 0.16	8.18 ± 0.37	0
	TAM	6	5.01 ± 1.20 ^c^	4.72 ± 1.08 ^c^	3.87 ± 0.99 ^c^	83.3^c^

CTL: control receiving corn oil as vehicle control; TAM: tamoxifen. ^a,b,c^: *p* < 0.05: treated vs. untreated.

## Data Availability

The data presented in this study are available in the article and Appendix A.

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
