# Peer review of "Potential Tamoxifen Repurposing to Combat Infections by Multidrug-Resistant Gram-Negative Bacilli"

_pharmaceuticals, 2021, doi:10.3390/ph14060507_

Round 1

Reviewer 1 Report

In their revised manuscript, Miró-Canturri et al. addressed all comments of the reviewers. Moreover, additional experiments were performed to show that tamoxifen treatment does not only prevent severer infections with the model strains but also can relieve an existing infection.

However, as two reviewers mentioned a major limitation of the proposed treatment strategy is the dosage of tamoxifen used. In their answer to reviewer 2, the authors mixed up mg ip/kg/d and mg po/d. A breast cancer patient with a body weight of 60 kg receives at highest a daily dosage of 0.67 mg/kg/d.  Ciprofloxacin dosage in mice 40 mg/kg/d and humans 16.6 mg/kg/d (1000 mg/d again in a slender person with 60 kg) is comparable, not at all “too below”.

I totally agree to the authors that treatment schedules are completely different in breast cancer and bacterial infections. To my knowledge there are no date available on short time exposure to high tamoxifen dosages. Maybe, the authors can add a respective sentence to the discussion.

Minor comments:

Lines 232f: type size

Line 411f: sentence needs revision

Author Response

In their revised manuscript, Miró-Canturri et al. addressed all comments of the reviewers. Moreover, additional experiments were performed to show that tamoxifen treatment does not only prevent severer infections with the model strains but also can relieve an existing infection.

Thank you for your constructive comments. Below you can find the answers to your specific comments. All changes in the revised manuscript are highlighted in red.

However, as two reviewers mentioned a major limitation of the proposed treatment strategy is the dosage of tamoxifen used. In their answer to reviewer 2, the authors mixed up mg ip/kg/d and mg po/d. A breast cancer patient with a body weight of 60 kg receives at highest a daily dosage of 0.67 mg/kg/d.  Ciprofloxacin dosage in mice 40 mg/kg/d and humans 16.6 mg/kg/d (1000 mg/d again in a slender person with 60 kg) is comparable, not at all “too below”.

The Reviewer is right. The different dosages in humans and mice are comparable and it is not correct to use “too below”. The Reviewer state in the next comment that there is no data about the treatment schedule of tamoxifen in bacterial infections in humans. To this regard we have included a new sentence in the revised manuscript (see lines (428-430).

I totally agree to the authors that treatment schedules are completely different in breast cancer and bacterial infections. To my knowledge there are no date available on short time exposure to high tamoxifen dosages. Maybe, the authors can add a respective sentence to the discussion.

As requested by the Reviewer, we have added a respective sentence to Discussion section (see lines 428-430).

Minor comments:

Lines 232f: type size

As requested by the Reviewer, we have corrected the type size (234-235).

Line 411f: sentence needs revision

As requested by the Reviewer, we have revised this sentence (see lines 413-415).

Reviewer 2 Report

In my opinion this version of the manuscript is publishable on Pharmaceuticals

Author Response

In my opinion this version of the manuscript is publishable on Pharmaceuticals

We thank the Reviewer for this comment and to accept our manuscript for publication in Pharmaceuticals.

Reviewer 3 Report

Potential tamoxifen repurposing to combat infections by multi-drug-resistant Gram-negative bacilli  by

Miró-Canturri et al. 

In this paper the authors provide the first evidence of an essential role played by tamoxifen in the regulation of immune cells’ traffic after bacterial infection.  The manuscript is well written and well proof-read.  The findings will make a significant advance in the field.  I have a few minor comments:

Page 1 line 22: In the abstract, please don’t use abbreviations which all people cannot understand.  Change “MCP-1” to “monocyte chemotactic protein-1 (MCP-1)”.  This will also increase visibility of the paper when people search for keywords.  This may not be necessary for IL and ERK, but it won’t do any harm.

Page 2 line 58: “increased by” Do you mean “increased in”?  Both can be right, depending on what the authors mean to say.

Page 2 line 59: “produces the mobilization”.  Please replace “produces” with an appropriate verb such as “causes” of “results in” etc. 

Page 2 line 62: Change “is” to “has been”

Page 3 line 84 and at other places: “the rates of these immune cells, from a total of 1.6 x 106 cells/mL” It is not clear what is meant by “rate”.  Cells/mL is a unit of concentration, not rate, which should have the unit of time included in it. 

Author Response

Potential tamoxifen repurposing to combat infections by multi-drug-resistant Gram-negative bacilli by Miró-Canturri et al. In this paper the authors provide the first evidence of an essential role played by tamoxifen in the regulation of immune cells’ traffic after bacterial infection.  The manuscript is well written and well proof-read.  The findings will make a significant advance in the field.  I have a few minor comments:

Thank you for your constructive comments. Below you can find the answers to your specific comments. All changes in the revised manuscript are highlighted in red.

Page 1 line 22: In the abstract, please don’t use abbreviations which all people cannot understand.  Change “MCP-1” to “monocyte chemotactic protein-1 (MCP-1)”. This will also increase visibility of the paper when people search for keywords. This may not be necessary for IL and ERK, but it won’t do any harm.

As requested by the Reviewer, We have detailed the abbreviations MCP-1, IL and ERK in the abstract section (see lines 22-24 and 61-62).

Page 2 line 58: “increased by” Do you mean “increased in”?  Both can be right, depending on what the authors mean to say.

The Reviewer is right. We have replaced “increased by” by “increased in” (see line 59).

Page 2 line 59: “produces the mobilization”. Please replace “produces” with an appropriate verb such as “causes” of “results in” etc.

As requested by the Reviewer, we have replaced “produces” by “results in” (see line 60).

Page 2 line 62: Change “is” to “has been”

As requested by the Reviewer, we have changed “is” to “has been” (see line 64).

Page 3 line 84 and at other places: “the rates of these immune cells, from a total of 1.6 x 106 cells/mL” It is not clear what is meant by “rate”.  Cells/mL is a unit of concentration, not rate, which should have the unit of time included in it.

As requested by the Reviewer and to avoid confusion, we have replaced “rate” by “percentage” (see lines 83 and 87).